# Epstein–Barr Virus Infection in Lung Cancer: Insights and Perspectives

**DOI:** 10.3390/pathogens11020132

**Published:** 2022-01-21

**Authors:** Julio C. Osorio, Rancés Blanco, Alejandro H. Corvalán, Juan P. Muñoz, Gloria M. Calaf, Francisco Aguayo

**Affiliations:** 1Population Registry of Cali, Department of Pathology, Universidad del Valle, Cali 760042, Colombia; cejulio704@gmail.com; 2Laboratorio de Oncovirología, Programa de Virología, Instituto de Ciencias Biomédicas (ICBM), Facultad de Medicina, Universidad de Chile, Santiago 8380000, Chile; rancesblanco1976@gmail.com; 3Advanced Center for Chronic Diseases (ACCDiS), Pontificia Universidad Católica de Chile, Santiago 8320000, Chile; acorvalan@uc.cl; 4Instituto de Alta Investigación, Universidad de Tarapacá, Arica 1000000, Chile; jp_182_mb@hotmail.com (J.P.M.); gmc24@cumc.columbia.edu (G.M.C.); 5Center for Radiological Research, Columbia University Medical Center, New York, NY 10032, USA; 6Universidad de Tarapacá, Arica 1000000, Chile

**Keywords:** Epstein–Barr virus, cancer, lung

## Abstract

Lung cancer (LC) is the leading cause of cancer death worldwide. Tobacco smoke is the most frequent risk factor etiologically associated with LC, although exposures to other environmental factors such as arsenic, radon or asbestos are also involved. Additionally, the involvement of some viral infections such as high-risk human papillomaviruses (HR-HPVs), Merkel cell polyomavirus (MCPyV), Jaagsiekte Sheep Retrovirus (JSRV), John Cunningham Virus (JCV), and Epstein–Barr virus (EBV) has been suggested in LC, though an etiological relationship has not yet been established. EBV is a ubiquitous gamma herpesvirus causing persistent infections and some lymphoid and epithelial tumors. Since EBV is heterogeneously detected in LCs from different parts of the world, in this review we address the epidemiological and experimental evidence of a potential role of EBV. Considering this evidence, we propose mechanisms potentially involved in EBV-associated lung carcinogenesis. Additional studies are warranted to dissect the role of EBV in this very frequent malignancy.

## 1. Introduction

Lung cancer (LC) is the leading cause of cancer death worldwide, with 2.21 million new cases and 1.80 million deaths in 2020 [1]. LC is a heterogeneous disease, broadly classified into two main groups: small cell lung carcinoma (SCLC), and non-small cell lung carcinoma (NSCLC) [2]. SCLC is a type of LC that shows a neuroendocrine origin, with a very aggressive behavior and poor prognosis [3]. NSCLC is the most prevalent type, including squamous cell carcinoma (SQC), adenocarcinoma (AdC), and large cell carcinoma (LCC) [4]. SQC shows a bronchial origin and is characterized by keratinization, intercellular bridges, tumor bulk, expression of cytokeratins 5/6, and diffuse expression of p63 [5]. AdC exhibits an alveolar origin and is characterized by thyroid transcription factor 1 (TTF1) and keratin 7 expression (KRT7) [6]. LCC is typically generated in the outer regions of the lungs and is characterized by KRT7 and epithelial membrane antigen (EMA) expressions, with a lack of TTF-1, p63, and keratin 5/6 expressions [7]. Tobacco smoke is the most frequent and well-known risk factor associated with LC [8]. Indeed, secondary metabolites from tobacco smoke promote mutations in genes such as TP53, KRAS, NRAS, HRAS, MYC, cyclin-dependent kinase inhibitor 2A (CDKN2A), and retinoblastoma (RB1) [9,10]. All of the aforementioned genes are involved in the increase in genome instability and LC progression [11]. However, additional environmental factors also play a role in LC development, such as radon, arsenic, and environmental pollution [12]. Tobacco smoke harbors more than 7000 chemicals [13] and approximately 70 of them are considered carcinogens [14], including polycyclic aromatic hydrocarbons (PAH), nitrosamines, acetaldehyde, arsenic, beryllium, cadmium, 1,3-butadiene, and azarenes, among many others [8]. Tobacco smoke chemicals can also be found inside the air we breathe as well as in our food and water [15]. In particular, large amounts of benzo[a]pyrene (BaP), a type of PAH, are released into the air through industry, agriculture, and emissions from combusting fossil fuels [16]. Other sources of BaP are produced when coal, oil, gas, wood, and garbage are burned [17].

Viral infections have been proposed to be involved in LC, namely, high-risk human papillomaviruses (HR-HPVs) [18], Merkel cell polyomavirus (MCPyV) [19], Jaagsiekte Sheep Retrovirus (JSRV) [20], John Cunningham Virus (JCV) [21], and Epstein–Barr virus (EBV) [22]. EBV is an exclusive human virus (without an animal reservoir) present in ~95% of the human population and establishing persistent infections during the lifetime [23]. EBV preferentially infects both B cells and epithelial cells [24] and is characterized as a class I carcinogen [25]. Epidemiological evidence suggests that EBV is present in a relatively small subset of LCs [22], with demographic variations, though an etiological relationship remains inconclusive. In this review, we address the epidemiological and mechanistic evidence regarding a potential role of EBV in this malignancy. Taking into consideration that previous reported data suggested a cooperation between EBV and xenobiotics for carcinogenesis [26], here we propose potential mechanisms that involve an interplay between environmental factors such as tobacco smoke and EBV for developing lung cancer. However, additional experimental and epidemiological studies are warranted to confirm these possibilities.

## 2. Epstein–Barr Virus: Structure, Replication, and Role in Cancer

### 2.1. Epstein–Barr Virus: Structure and Replication Cycle

EBV is an enveloped gamma herpesvirus and a member of the *Lymphocryptovirus* genus, subfamily Gammaherpesvirinae [27]. The EBV core contains a double-stranded linear DNA virus with approximately 172-kpb, enclosed by a 162-capsomer icosahedral capsid (Figure 1). The core is surrounded by the tegument, which is composed of enzymes and proteins involved in viral replication [28]. Primary EBV infection during childhood is frequently asymptomatic, though infection during adolescence can lead to symptomatic infectious mononucleosis (IM) [29]. After EBV entry in the Waldeyer’s ring cells, EBV infects naïve B cells, leading to the establishment of a latent infection in memory B cells that then enter the peripheral circulation [30]. It is not clear if EBV replicates in epithelial cells from the Waldeyer’s ring before B-cell infection. Eventually, during EBV reactivation, the virus can re-enter Waldeyer’s ring structures for transmission through saliva [31]. EBV gH/gL and gB glycoproteins are involved in the viral entry into epithelial cells by the fusion machinery [32], while glycoprotein 42 (gp42) is only required during infection of B cells [33]. Additionally, depending on the CR2 (Complement receptor type 2, CD21) expression levels, gp350 can be used as a ligand for attachment to CD21(+) epithelial cells or the gH/gL complex for attachment to CD21 (−) epithelial cells [34]. Importantly, the Ephrin receptor tyrosine kinase A2 (EphA2) has been recently proposed to be the epithelial EBV receptor [35]. After viral attachment, entry occurs by cell fusion, allowing the entry of the tegumented capsid into the cytoplasm [36]. Next, the capsid with the genome is transported to the periphery of the nucleus [37] and the viral genome crosses the nuclear pore complexes to enter the nucleus [38]. The Epstein–Barr nuclear antigen 1 (EBNA1) gene is expressed, leading to the EBV genome tethering to the host genome and is maintained as an episome [39,40]. During lytic cycle activation, the immediate-early (IE) proteins are first expressed. Indeed, the IE Zta and Rta proteins act as both key transactivators and master regulators of the lytic cycle allowing the expression of the viral DNA polymerase (BALF5) for viral DNA replication by a rolling circle mechanism [41]. Indeed, the recruitment and assembly of core replication proteins at the oriLyt are a requisite for EBV genome replication. The involved viral proteins are BALF5, BMRF1, BALF2, BBLF4 BSLF1, and BBLF2/3. The synthetized linear EBV genome is cleaved at the terminal repeats and packaged into viral capsids. The serine/threonine-protein kinase BFRF1 and BFLF2 genes allow intranuclear nucleocapsids to move towards the inner leaflet of the nuclear membrane and sprout from the nuclear envelope [38]. Next, the access of nucleocapsids to the nuclear membrane is mediated by the viral BGLF4 protein [38]. The nucleocapsids can then be transported into the juxtanuclear viral assembly compartment. Finally, the virions are released from the infected cells by exocytosis [42].

### 2.2. Role of EBV in Epithelial Cancers

EBV is the causal agent of undifferentiated nasopharyngeal carcinoma (NPC) [43] and a subset of gastric carcinomas (GCs) [44]. Additionally, EBV has been detected in other tumors such as those arising in the breast [45], colon [46], prostate [47], and lung, though an etiological association has not yet been established. Knowledge of the role of EBV in carcinomas has arisen from studying this infection in NPC and GC. Indeed, undifferentiated NPC is a recognized model of EBV-driven epithelial carcinogenesis [48]. Type II latency is the predominant form of latency found in undifferentiated NPC [48], in which LMP1, LMP2A, EBNA1, BARF1, EBER, BARTS, and BamHI-H rightward fragment 1-derived miRNAs (BHRF1) are detected [43]. The BHRF1 miRNA cluster encodes four miRNAs, and BART generates 28 miRNAs encoded by two miRNA clusters [49]. Viral miRNAs can regulate viral gene expression related to cancer development and progression [50].

Interestingly, evidence suggests that these latent EBV-expressed products show oncogenic properties [30]. Indeed, EBV oncoproteins activate signaling pathways such as Wnt/β-catenin, NF-κB, JNK, JAK/STAT, EGFR/MAPK, and PI3K/AKT (Figure 2) [51]. Regarding GCs, a recent meta-analysis reported that EBV prevalence among patients with GC was 8.77% worldwide [52]. The mechanisms by which EBV is involved in GC are not clear, but it is suggested that an EBV latency II profile promotes changes in the host genome methylation pattern, with subsequent increases in genomic instability [53]. Since latent EBV infection is generally not found in healthy nasopharyngeal tissue, it is plausible that previous genomic alterations are a condition for establishing EBV latency [54]. Importantly, both NPCs and GCs show genetic alterations (germ, somatic, and environmental mutations) that can predispose the tissue to latent EBV infection [55]. Regardless of the type of genetic alteration, CDKN2A (p16) loss and cyclin D1 dysregulation are key alterations that increase the genetic instability of cells. P16 is a negative regulator of cyclin D1 signaling, and its blocking or deregulation allows the progression of the cell cycle from the G1/S transition, which is related to increased cell proliferation, genomic instability, and cancer progression [56]. Increased cell proliferation due to the ectopic expression of cyclin D1 has been evaluated in tissues that have reported high efficiency in supporting latent EBV infection mediated by enhanced EBNA1 expression [57]. EBNA3C is related to cyclin A and drives a hyperphosphorylated form of retinoblastoma protein (Rb), allowing S phase entry. EBNA3 is also involved in Rb and p27 ubiquitination and subsequent degradation, which facilitates G1/S phase transition [58]. Additional genetic alterations, epigenetic modifications, and cofactors such as chronic inflammation, immunosuppression, and environmental mutagens are also required for this multistep oncogenic process.

### 2.3. EBV Latency and Lytic Cycle in Cancer

The establishment of EBV latency in epithelial tissue is a critical hit during EBV-driven carcinogenesis [59]. It is well-known that during latency, the EBV genome remains within the nucleus in an episomal form, and this is maintained at a constant number of copies after cell division [60]. This event occurs in B cells and in tumor epithelial cells, although evidence shows that latency is not established in normal epithelial cells, which per default activate the lytic cycle [61]. However, in EBV-driven NPCs and EBV-associated GCs (EBVaGCs), the EBV genome is always found in latency. Four latency forms (latency 0, I, II, and III) have been defined based on the pattern of expressed EBV products [62]. Latency 0 occurs in resting memory B cells, with only EBERs being detected [59]. Latency I is detected in Burkitt’s lymphoma (BL) with only EBERs and EBNA1 expression. Latency II is found in Hodgkin’s disease (HD), GC, and NPC with EBERs, BARTs, EBNA1, and three LMP genes being expressed [63]. Latency III is the least restricted latency type and is found in immortalized lymphoblastoid cell lines with EBERs, BARTs, six EBNA, and the three LMPs genes being expressed. It has been hypothesized that previous cell alterations, including p16 loss and cyclin D1 overexpression, are necessary for EBV latency establishment during NPC development [57]. However, the mechanisms and factors involved in latency establishment in epithelial cells are unknown.

Although latency establishment is a requisite for EBV-driven cancer, evidence shows that lytic cycle activation is important during EBV-driven carcinogenesis as well. Indeed, the abortive lytic cycle (ALC) involves the expression of latent genes and a restricted set of lytic genes, though without EBV maturation [64]. This is based on diverse studies in which some lytic proteins are consistently detected in cancers including Zta, Rta, and BARF1, among others [65]. In particular, BARF1 is an oncogenic early EBV protein with functions in immune evasion [66]. In latently infected B cells, lytic EBV promoters are methylated while they remain hypomethylated in normal oral keratinocytes (NOK) and Akata cells [67]. Additionally, EBV-infected human telomerase-immortalized NOK have shown CpG island hypermethylation in the transcription factor lymphoid enhancer factor 1 (LEF1) promoter region, which explains the metastatic phenotype of EBV-associated carcinomas [68]. Latent EBV infection causes genome-wide DNA methylation in GC through increased expression of LMP1 on DNA methyltransferases [69]. Global DNA hypermethylation of both viral and host DNA is frequently found in EBVaGCs [70]. The Zta protein preferentially binds to CpG-methylated motifs to reverse epigenetic silencing of viral DNA, leading to lytic activation [71].

## 3. EBV Infection in Lung Cancer

### 3.1. Pathogenesis of Lung Cancer

The LC pathogenesis is characterized by the accumulation of diverse genetic abnormalities, including chromosomal instability (CIN), loss or gain of whole chromosomes (aneuploidy), loss of heterozygosity (LOH), chromosomal deletions and rearrangements, as well as loss-of-function mutations in TSGs while activating point mutations of oncogenes. Additionally, epigenetic alterations including aberrant DNA-methylation patterns, among others, are involved in LC development (reviewed in [72,73]). The occurrence of CIN during cell division, which leads to other chromosomal aberrations, is considered one of the most important events associated with cancer development [74]. In bronchial epithelium, aneuploidy increased from moderate to severe dysplasia and carcinoma in situ lesions and was also associated with smoking status [75,76]. For instance, the percentage of cells displaying gain of whole chromosome 7 was reported to be increased from preinvasive to invasive lung lesions [75]. Chromosome 7 contains some oncogenes whose function is related to cell division, including the epidermal growth factor receptor (EGFR) (7p11.2) and MET (7q31.2) [77]. In addition, aberrations in the number of whole chromosomes have been evidenced in 51.8% to 84.8% of NSCLCs, suggesting a crucial role in LC progression [78,79,80]. In fact, a meta-analysis conducted by Choma et al., 2001 demonstrated the association of aneuploidy with poor prognosis of NSCLC patients [81]. LOH was found at chromosomes containing TSGs such as 17p13 (TP53), 9p21 (CDKN2A), and 3p14 (FHIT) in bronchial epithelium of smokers [82]. An increased LOH at chromosomes 3p21, 5q21, and 9p21 was also detected in normal bronchial mucosa adjacent to NSCLC compared to normal bronchial cells obtained from non-cancer patients [83]. Moreover, LOH was significantly related to the occurrence of carcinoma in situ [84]. Chromosomal rearrangements such as echinoderm microtubule-associated protein-like 4 (EML4) and anaplastic lymphoma kinase (ALK) fusion gene represent an important genetic alteration in LC. The aberrant fusion of the N-terminal domain of EML4 and the C-terminal region of ALK encodes a cytoplasmic chimeric protein with constitutive tyrosine kinase activity [85]. The expression of the EML4-ALK fusion gene is more frequent in lung AdCs and non-smoker patients, as well as being exclusive to the K-ras and EGFR mutations [86]. The EML4-ALK fusion protein increased the phosphorylation of STAT3 and AKT in human bronchial cells [87]. Increased tumorigenic properties in vivo of mouse 3T3 fibroblasts expressing EML4-ALK fusion protein were evidenced when compared with EML4- or ALK-transfected cells [85]. Transfection of human bronchial cells with the EML4-ALK fusion gene resulted in increased cell proliferation rates and enhanced anchorage-independent growth, though an absence of tumor formation was observed in vivo [87].

Mutations in TP53 are more frequent in SCLCs and SQCs compared to AdCs [88]. A high frequency of deletions was found in the chromosomal region 17p13, which contains the TP53 gene, in lung tumors [89]. In addition, TP53 mutations occur in about 50% of NSCLCs, mostly characterized by GC > TA transversions [90]. Loss of function of p16 was evidenced in NSCLCs, including promoter hypermethylation, point mutations, and loss of heterozygosity in the 9p21 region [91]. A low expression level was reported as well as CDKN2A gene promoter hypermethylation in alveolar atypical hyperplasia, the precursor lesion of lung AdC [92,93]. Loss of RB expression was significantly increased in SCLCs when compared with NSCLCs [94]. RB knockdown using shRNA vectors increased the proliferation rates of NSCLC cells as well as tumor formation in vivo [95]. Moreover, the somatic inactivation of both RB1 and P53 alleles in mouse lung epithelial cells led to neuroendocrine tumor formation, resembling the human SCLC [96]. Interestingly, loss of RB was associated with the EGFR mutant NSCLC that transformed into SCLC, the most aggressive form of LC disease [97]. Wild-type EGFR was found overexpressed in more than 60% of NSCLCs [98,99] and was associated with a decreased overall survival and chemoradiotherapy resistance [100]. On the other hand, EGFR mutations in the tyrosine kinase domain are frequently detected in NSCLCs. In a meta-analysis conducted by Zhang et al., 2018, EGFR mutations in NSCLCs ranged from 9.6 to 82.2% [101]. Another oncogene with a role in LC progression is the human epidermal growth factor receptor 2 (HER2). The expression of HER2 was detected in approximately 25% of LCs by immunohistochemistry (IHC) and 20% of LCs by fluorescence in situ hybridization (FISH) [102]. In addition, HER2 mutations were more frequently found in AdC (3%), females, and never smokers [103]. The most common mutation in the HER2 kinase domain is a 12 bp insertion in exon 20 (YVMA) [104,105]. HER2^YVMA^ expression results in constitutive phosphorylation of HER2 and EGFR activation in the presence of tyrosine kinase inhibitors (TKIs). HER2 mutant also displayed an increased capacity to transform bronchial epithelial cells when compared to wild-type HER2 [106]. HER2 aberrant expression is associated with a decreased overall survival of NSCLC, mainly with AdC histology [102]. K-ras mutations are also observed in about 18–40% of lung AdCs [107,108,109] and are less common in SQCs [107]. It was demonstrated that K-ras mutations led to increased cell migration of immortalized airway cells through Akt activation [110]. Moreover, K-ras mutations were found to be related to the progression from precursor lesions (e.g., alveolar atypical hyperplasia and adenoma) to lung AdC in mice [111], which suggests a role in the development of these tumors. K-ras, EGFR, and HER2 mutations are mutually exclusive [105,112]. MYC family oncogenes (c-myc, N-myc, and L-myc) are commonly amplified in LC subtypes: AdC, SQC, and SCLC [113]. Approximately 50% of NSCLCs present activator changes in the mitogenic cascade (RAS-RAF-MEK-ERK) that induces c-MYC [114]. For instance, c-myc expression was evidenced in 35.7–58.6% of NSCLCs although an increased expression was reported in SQC compared with AdC [115,116,117]. Myc amplification and overexpression have been observed in SCLC [118]. In surgically resected SCLC, L-myc and c-myc protein expressions were detected in 6.5 and 8.7% of samples, respectively [119]. MYC family suppression induced cell cycle arrest and apoptosis inhibition in SCLC cells displaying both TP53 and RB1 inactivation [120]. Moreover, MYC inhibition induced tumor regression in K-Ras-driven LC in mice as well as in p53-deficient animals [121], which suggests the contribution of MYC for the progression of these tumors. hTERT expression has been reported (33% to 94%) in NSCLCs with poorer overall survival at three years [122] and it was significantly higher in NSCLC when compared with non-tumor tissues [123]. Among NSCLCs, AdCs display an increased expression of hTERT when compared with SQCs [122]. Furthermore, hTERT mRNA expression increased from atypical adenomatous hyperplasia to bronchioloalveolar carcinoma [124]. hTERT has been correlated with lymph node metastasis, tumor cell differentiation, and LC aggressiveness [125]. The expression of hTERT was also related to both decreased overall survival and disease-free survival of NSCLC patients [126,127]. hTERT has been associated with changes in EGFR expression that can affect NSCLC proliferation, and hTERT promoter methylation can enhance SCLC progression and metastasis [128].

### 3.2. Epidemiology of EBV in Lung Cancer

The relationship between LC and EBV, analyzed in diverse studies, remains unclear. Indeed, EBV is detected at highly variable frequencies in clinical specimens of LC. These variations can be explained by sociodemographic factors, type of clinical specimens (i.e., paraffin embedded tissues, fresh tissue, etc.), histological type of LC, and even the sensitivity and specificity of the molecular assays (i.e., polymerase chain reaction (PCR), ISH, IHC). It is important to point out that EBER ISH is the gold standard method with the capacity to detect and localize latent EBV in tissue samples [129]. That said, other EBV detection methods such as LMP1 IHC are commonly used, despite the fact that this protein is often undetectable even when EBER ISH is clearly positive [130]. IHC is another technique reported to detect targets such as EBNA1, EBNA2, LMP2A, BARF1, and BZLF1 (Zta). Zta is a useful marker of lytic viral replication [131], and BARF1 can be used as a marker of epithelial malignancies [132]. Additionally, PCR-derived methods and nucleic acid sequence-based amplification (NASBA) assays are also used at the research level but rarely in clinical laboratories [130].

Pulmonary lymphoepithelioma-like carcinoma (PLELC) is a rare form of NSCLC closely related to EBV infection. PLELC is characterized by an undifferentiated carcinoma accompanied by an intense lymphocytic infiltrate, which histologically resembles NPC [133]. The genomic landscape of PLELC differs from the most frequent NSCLC subtypes but shows similarity with EBV-driven NPCs [134]. These tumors are most frequent in the Asian population [135], which is also an endemic area of NPC. In southern China, EBV was reported in 30 (93.8%) of 32 LELCs by EBER ISH. Furthermore, 53.3% of EBER positive specimens expressed LMP1, and 23.3% also expressed the viral capsid antigen (VCA) [136]. Other studies conducted in China and Taiwan found EBV positivity in 100% of pulmonary LELC [137,138]. The expression of the anti-apoptotic Bcl-2 was significantly increased in five EBV-positive PLELCs when compared with other lung tumors such as SQC, AdC, and LCC [137]. In addition, 91 lung LELCs from China associated with EBV showed molecular alterations in NF-κB, JAK/STAT, and cell cycle signaling pathways [138]. Recently, another type of EBV-related NSCLC was described, displaying low lymphocytic infiltration, but similar patient features and molecular profile to classic PLELC [139,140]. Overall, the presence of EBV was significantly increased in PLELC when compared with non-PLELC [136,137]. Among the different studies carried out to detect EBV in other LCs, we can highlight the following: In North China, EBV was detected in 33.3% of LC patients [141]. In southern China, EBV was reported in 30 (58.8%) of 51 lymphoepithelioma-like carcinomas (LELCs) by EBER ISH. Furthermore, 56% of EBER positive specimens expressed LMPs, and 23% also expressed the viral capsid antigen (VCA) [136]. In China (Henan), 108 cases of LC were evaluated by EBER ISH. Additionally, expressions of LMP1 and Bcl-2 were evaluated by IHC. EBV was reported in 33.3% (EBER positive) and LMP1 was expressed in 6%. Bcl-2 was higher in EBV (+) LC than that in EBV (−) [142]. In (Beijing) China, EBV was detected in 37.9% of 87 SQCs by IHC [143]. In Hong Kong, EBV was reported in 5.4% of 167 NSCLCs by EBER ISH [144]. In Taiwan, EBV was detected in 11 (8.2%) of the 127 NSCLC cases by EBER1 ISH [137]. In Japan, 81 LCs were examined with five cases being EBER1 positive (three SQC, and two AdC) [145]. In Spain, 19 AdC and SQC cases were evaluated by EBER ISH, and 12 (63.2%) were positive [146]. Other studies have used molecular methods such as PCR or qPCR for EBV detection in LCs. In Iran, EBV was detected in 10.4% of 48 LCs by using an EBV-Eph PCR kit and IHC for LMP1 [147]. In India, EBV was detected in 2.76% of 75 cytological specimens from LC patients [148]. In Italy, EBV was detected in 26% of 65 (21/65) NSCLCs and 20% of five (1/5) SCLC in exhaled breath condensate by using real-time PCR assay [149]. In China (Wuhan), EBV DNA was detected by PCR in 52% of 48 LC fresh tissues [150].

In recent years, new technologies such as next-generation sequencing (NGS), whole exome sequencing (WES), and microRNA (miRNA) analysis have been used to characterize EBV presence in LC. Accordingly, EBV was detected in 6.1% (4/66) of NSCLCs. Additionally, the LC strains were classified as V-val and China 1 subtypes according to the amino acid sequence of EBNA1 and LMP1 proteins, respectively [151]. In addition, 91 lung LELCs from China associated with EBV showed molecular alterations in NF-κB, JAK/STAT, and cell cycle signaling pathways [138]. In the United States, data sets from 1017 LCs and 110 paired adjacent normal lung specimens revealed EBV transcripts in three LCs and one normal lung sample. In the sample with the highest EBV coverage, BamHI-A rightward transcripts (BARTs) accounted for the most EBV reads by RNA-seq. Additionally, EBNA1, LMP1 and LMP2 expression was observed. Interestingly, several viral circular RNA candidates were also detected. Thus, this study reported for the first time a type II latency-like viral transcriptome in LC in vivo [152]. Interestingly, expression of lytic genes was also detected. Among them, BZLF1 transcripts, encoding for the Zta protein, were also found in EBV-positive cases, suggesting the importance of lytic activation in EBV-driven tumors, as previously mentioned in chapter 2.3 [152,153]. Conversely, other studies have shown the absence of EBV in different histological types of LC. Indeed, a study conducted in lung AdC and mesotheliomas did not detect EBV by ISH [154]. In another study, 23 Asian SCLCs were analyzed by EBER-ISH without detecting EBV gene expression. In this study, eight SCLCs showed focal positivity for EBNA1 by IHC with only one case positive for LMP-1 by PCR [155]. Another study in Singapore showed no detection of EBV by ISH in 110 lung AdCs [156]. In 2012, the first analysis of EBV miRNAs in LC was carried out in 290 cases. Differences in EBV miRNAs between AdCs and SQCs were determined by microarrays (*p* < 0.01 for nine out of 16 EBV miRNAs), though such differences were not found for BART1, BART2, and BHRF1–3 expressions (*p* = 0.53, 0.94, and 0.47, respectively) [157]. In Korea (Busan), EBV-miR-BART19 upregulation was detected in four AdC samples by microarrays, and the results were validated by qPCR and ISH [158]. In (Shandong) China, EBV-miR-BART14-5p downregulation was detected by miRNA microarrays in plasma from patients with NSCLC harboring EGFR-activating mutations [159].

In France, 118 LCs were analyzed by EBER1-ISH without detecting EBV [160]. Taken together, although EBV has been reported to be present in a subset of LCs from some populations, the role of EBV and its mechanisms in LC have not yet been established (Table 1).

### 3.3. Potential EBV-Mediated Lung Cancer Mechanisms

#### 3.3.1. Interactions with Tobacco Components

##### Tobacco Smoke May Facilitate EBV Latency Establishment in Lung Epithelial Cells

Normal epithelial cells are assumed to be unable to sustain a latent EBV infection, which is a requisite for EBV-driven cancer [54]. However, tumor epithelial cells can support EBV latency as demonstrated in NPC and EBVaGC, where EBV is generally found in a latency II status, as previously stated [48]. Thus, it is plausible that previous cell alterations are a required condition (or pre-condition) that allows EBV latency establishment in epithelial cells [61]. Tsang et al. proposed that cell alterations including p16 loss and Cyclin D1 overexpression are among the alterations necessary for EBV latency in nasopharyngeal cells, leading to NPC [61]. Indeed, in vivo experiments have determined that p16 expression is a major barrier to proliferation during EBV-driven B-cell transformation [161]. Cyclin D1 gene dysregulation has been detected in smoking patients and could be associated with the spread of preneoplastic clones across the bronchial epithelial surface [162]. Dysregulated expression of cyclin D1 has been reported in association with persistent EBV infection in NPC [57]. This notion may be applicable to other models of cancer in which EBV plays a potential oncogenic role, including LC. The fact that tobacco smoke components can provide such alterations, including p16 loss [163] and cyclin D1 overexpression [164], suggests that EBV latency establishment may be favored in the lung cells of smoker subjects. This possibility is compatible with the model established by Becnel et al., who proposed that pro-tumor signals such as chronic inflammation result in dysplastic lesions, supporting EBV infection and maintenance of type II latency in the lung epithelia [22]. Considering that tobacco smoke is a frequent cause of the chronic obstructive pulmonary disease (COPD) that is caused by chronic inflammation [165], we suggest that tobacco smoke is a factor potentially leading to EBV latency establishment in lung cells. Of note, we consider that independent of chronic inflammation, normal lung epithelial cells are susceptible to EBV infection, though this is per default lytic. Thus, the EBV latency establishment probably requires the involvement of cofactors in lung epithelial cells. However, additional studies are necessary to confirm these possibilities.

##### Tobacco Smoke Impairs Immune Response Facilitating EBV Infection in Lung Epithelial Cells

Studies have shown that tobacco smoke alters the immune responses in epithelial lung diseases and can induce activation of an inflammatory response [166]. In addition, tobacco smoking can increase the frequency of memory B cells and reduce the production of IgA, IgG, and IgM [167]. Furthermore, it is possible for tobacco smoke to affect different types of lymphocytes such as T helper cells and regulatory T cells. It is possible for tobacco smoke to affect B lymphocyte development through down-regulation of transcription factor 3 (E2A immunoglobulin enhancer-binding factors E12/E47) (TCF3), which is associated with a decline in the percentage of memory B cells [168]. Additionally, experimental data suggested that tobacco smoking reduces the transporter associated with antigen processing 1 (TAP1) protein abundance and HLA class I levels. This immune change can promote subsequent malignant cell transformation [169]. Loss of the antigen-presenting molecules (including both MHC class I alleles and TAP-1) is frequent in LC [170]. Tobacco smoke also leads to viral lung infections with an increased incidence and severity due to airway epithelium oxidative damage [171]. Furthermore, tobacco smoking increases the oxidative stress that has the potential to promote the maintenance of viral genomes and EBV lytic reactivation [26]. EBV-positive BL cell lines studies have shown an increased ROS expression in both type I and III latency [172]. Additionally, chemicals in tobacco smoke induce activation of NF-κB and ROS pathways, which in turn promote chronic inflammation, autoimmune damage, and cancer progression [173]. In comparison, LMP1 induces NF-κB activation, increasing its transcriptional activity in EBV-infected MOLT4-DL cells [174]. Importantly, NF-κB activation suppresses the expression of DNA damage-binding protein (DDB1), which increases genomic instability in epithelial cells [43,175]. The EBNA gene activates NF-κB to drive oncogene expression (MYC, BCL2) and increases resistance to apoptosis in lymphoblastoid cell lines [176]. In addition, EBNA1 promotes DNA damage, inducing ROS pathways (SOD1, Prx1) in NPC cells [177]. The change in the host’s immune response can create pressure for EBV to be pushed into latency, although over time EBV can reactivate [178,179]. Pressure on the immune system can be caused by metabolic changes in memory B cells after EBV infection due to the increase in glucose uptake and the activation of immunometabolic functions [180]. LMP1 expression can affect glycolysis due to the energy increase for NPC pathogenesis and progression. Indeed, LMP1-mediated metastasis involves IGF1-mTORC2 signaling activation and nuclear acetylation of the Snail promoter by the pyruvate dehydrogenase E1 component subunit alpha (PDHE1α) [181]. EBV is a down-modulator of the human leukocyte antigen (HLA) class I pathway with a close relationship to impaired glucose uptake. Studies have shown that EBVaGC presents extreme DNA hypermethylation [182], and tobacco smoke increases the aberrant methylation in CpG islands [183]. Some genetic alterations in cellular genes such as indoleamine 2,3-dioxygenase (IDO), PD-L1, cytotoxic T-lymphocyte–associated antigen 4 (CTLA-4), and V-domain immunoglobulin suppressor of T cell activation (VISTA) have been implicated as supporting EBV infection in the lung epithelia [152], and VISTA is a new target for the immunotherapies against cancer [184]. Tobacco smoke causes lowered IDO activity with immunosuppression [185], and EBV infection increases the expression level of IDO and other immune molecules such as tumor necrosis factor alpha (TNF-α) and interleukin-6 (IL-6) [186]. Clinical assays have shown that enhanced IDO activity is associated with poor prognosis in patients with LC [187]. Additionally, tobacco smoke can suppress the normal function of B cells and change the standard immune functions [168,188].

##### Tobacco Smoke Can Activate the EBV Lytic Switch

Epidemiological evidence suggests that tobacco smoke is associated with EBV lytic cycle activation [189]. Furthermore, a dose–response relationship with EBNA1-IgA and Zta-IgA seropositivity [190] can be used as a predictive biomarker of NPC [191]. Thus, tobacco smoke can promote the activation of the EBV lytic cycle, which can spread the virus from B cells to epithelial cells. This effect of tobacco smoke may occur through the promotion of oxidative stress, hypoxia, and inflammation [41,192]. EBV reactivation by tobacco smoke components has been shown in Akata and B95-8 lymphoblastic cell lines [191] and can be mediated by reactive oxygen signaling and high oxidative stress [193]. Additionally, EBV detection is 2.4 times more frequent in current smokers with GC when compared to non-smokers [194]. Interestingly, tobacco smoke and BaP induce PD-1/CTLA-4 in lung epithelial cells [195,196] and can block the adaptive immune response [195]. PD-1/CTLA-4 can affect the latent/lytic proportion of EBV-infected B cells [197]. 

#### 3.3.2. Lung Cancer Cells May Facilitate EBV Entry through Increased CD21 Expression

CD21 expression facilitates EBV infection, and experimental data demonstrated that ectopic CD21 expression increases EBV infection [198]. Additionally, a low level of CD21 expression was considered sufficient for EBV infection [199]. Taking these findings into consideration, we evaluated the relationship between the CD21 (CR2) transcript expression in 1325 lung cancers vs. normal tissue. CD21 differences between SQC and AdC were also compared. This bioinformatic approach showed that CD21 expression was increased 1.8-fold in malignant tumors compared to normal tissue and increased 1.3-fold in lung AdC when compared to lung SQC (Figure 3). These analyses suggest the possibility that lung tumor cells expressing CD21 may favor EBV entry, although experimental confirmation is necessary.

#### 3.3.3. EBV Interactions with other Environmental Contaminants in Lung Cancer

Airborne pollutants have been etiologically related to LC. Indeed, it has been demonstrated that PM2.5 exposure is associated with inflammation through transforming the growth factor-β/Smad signaling pathway with increased risk for lung AdC [201]. Even at a low concentration, PM2.5 was significantly correlated with the incidence of SQC and AdC [202]. Importantly, arsenic trioxide (As2O3) is another environmental contaminant (present in water, air, and food) related to LC that can activate EBV lytic gene expression in epithelial cells [203]. However, As2O3 is also able to decrease cell proliferation, promote apoptosis, and exhibit anti-angiogenesis properties in the lungs, inhibiting tumor metastasis [204]. Radon is an environmental carcinogen and the first risk factor for LC in nonsmokers [205]. Radioactive radon particles that can become lodged in the lung tissue and eventually lead to LC [206]. Radon exposure can lead to pulmonary inflammation, and some IL variants in never smokers have shown an increased LC risk [207]. Interestingly, EBV DNA and RNA sequences have been reported in SQC and AdC with exposure to radon [146]. Other environmental factors that potentially interact with EBV include pesticide/herbicides. Among these, chlorpyrifos (CPF) stands out since it can interact with EBV through the activation of oxidative stress (OS). Furthermore, CPF can increase EBV early antigens (EA) and promote lytic infection. CPF is a type of organophosphate that can increase leukemia, lymphohematopoietic, brain, colorectal, lung, and breast cancer risk [208]. It has been reported that polychlorinated biphenyls (PCBs) are immunosuppressive agents that increase NHL risk [209]. Additionally, new evidence shows that PCBs and persistent organic pollutants (POP) are associated with LC risk in the general population, even at low doses [210,211]. To investigate the associations between pesticide exposures and HL risk, three population-based studies determined that terbufos (organophosphate) was significantly associated with HL (OR: 2.53, 95% CI 1.04–6.17) (95). Additionally, occupational terbufos exposure was associated with LC, leukemia, and non-Hodgkin lymphoma [212]. Other chemicals that activated the EBV lytic cycle have been discovered from food metabolism. One of these chemicals is 2,3,7,8-tetrachlorodibenzo-p-dioxin (TCDD), which can interact with B cells through aryl hydrocarbon receptor (AHR). The latter, in turn, can directly interact with EBNA-3, affecting viral latency [213]. The activation of the EBV lytic cycle by TCDD leads to a new viral infection and EBV-associated cellular transformation; it also induces EBV reactivation in both activated B cells and salivary epithelial cells [214]. the AHR activator TCDD boosts interleukin 1 (IL-1), IL-6, and IL-8 gene expression through the NF-κB pathway in NSCLC patients [215]. Other chemical carcinogens that can act synergistically with EBV are volatile N-nitrosamines. There are reports of contamination of Chinese salted fish with volatile N-nitrosamines, resulting in increased risk of developing NPC [216] and LC [217]. A hypothetical mechanism of EBV-tobacco/environmental contaminant interaction is shown in Figure 4. 

## 4. Conclusions and Remarks

LC is a highly prevalent malignancy worldwide with tobacco smoke being the most important risk factor. Additional risk factors are environmental pollution, radon, and arsenic, among others. The etiological role of viral infections such as EBV has not been established, although epidemiological evidence suggests that this virus may be involved in a very restricted subset of cases. Considering the evidence established in lung and additional models of epithelial cancers, we suggest that EBV can cooperate with tobacco smoke at different levels for lung carcinogenesis. The identification of clonal EBV in LC specimens will contribute to elucidation whether the infection occurs early during the neoplastic development. Thus, further knowledge of these mechanisms is necessary for the development of new therapies to improve the treatments and the diagnosis of diseases related to EBV biology.

## Figures and Tables

**Figure 1 pathogens-11-00132-f001:**
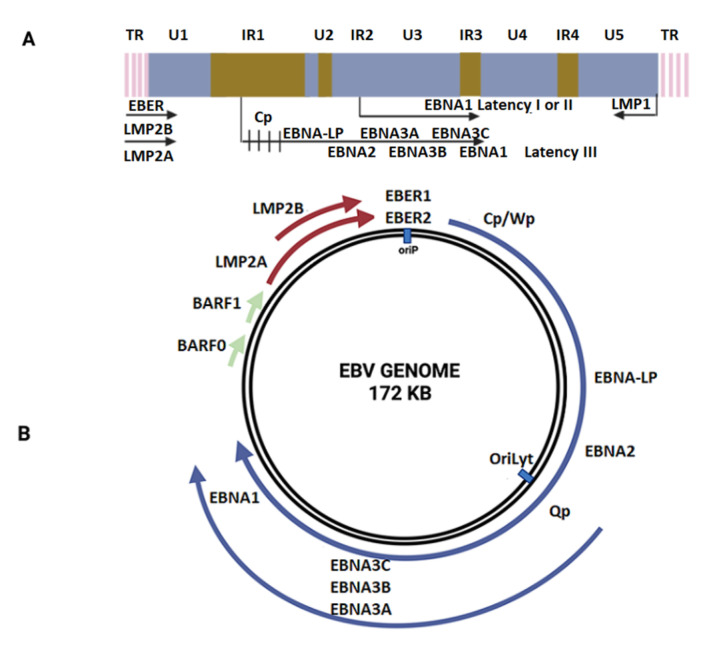
Epstein–Barr virus genome organization. (**A**) EBV lineal genome. (**B**) EBV circular genome representation.

**Figure 2 pathogens-11-00132-f002:**
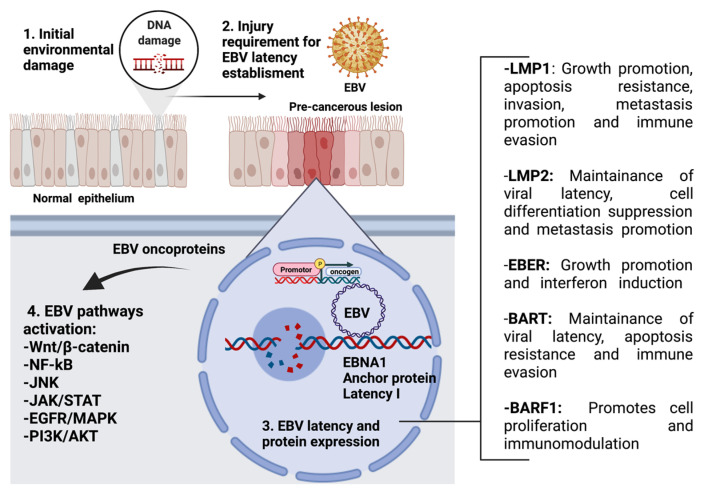
EBV proteins in epithelial tumors and their functions in cancer progression. This model suggests initial damage that can be caused by environmental pollutants (xenobiotics). Next, the epithelial injury allows the EBV latency establishment with the subsequent EBV protein expression. Finally, the EBV protein expression allows the activation of different cell signaling pathways such as Wnt/β-catenin, NF-κB, JNK, JAK/STAT, EGFR/MAPK, and PI3K/AKT.

**Figure 3 pathogens-11-00132-f003:**
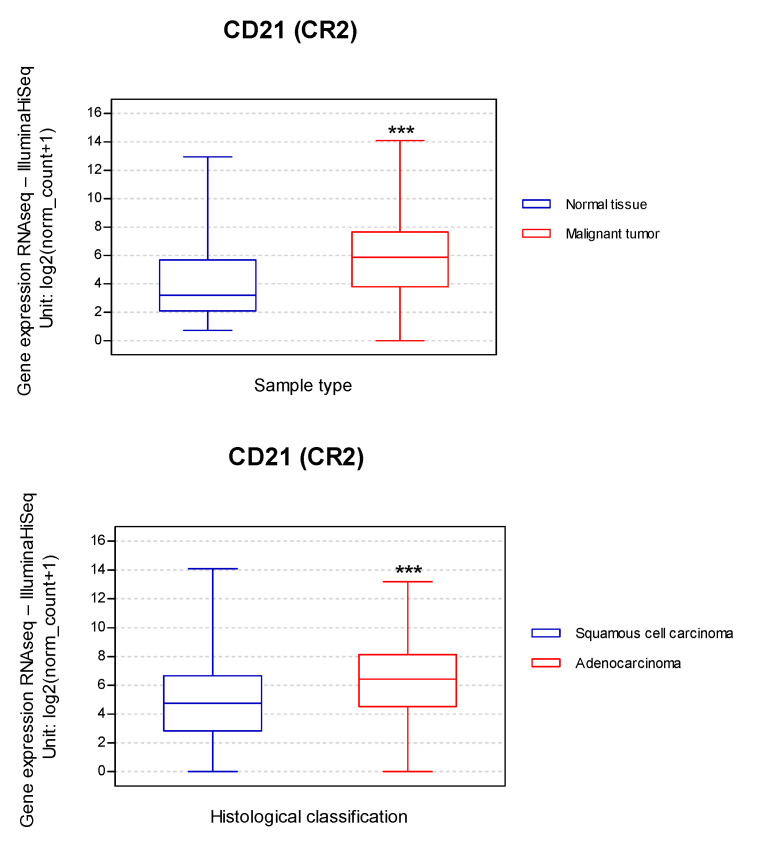
CD21 (CR2) transcript expression in LC (TCGA, n = 1325) according to sample type and histological classification of tumors (***: *p* < 0.001 for both; Welch’s *t*-test). The expression of CD21 was increased 1.8-fold in malignant tumors (1323 primary tumors and two recurrences) and increased 1.3-fold in lung AdC compared to normal tissue and lung SQC. Raw data were extracted from University of California, Santa Cruz (ena.ucsc.edu). UCSC Xena functional genomics explorer (https://xenabrowser.net) [200].

**Figure 4 pathogens-11-00132-f004:**
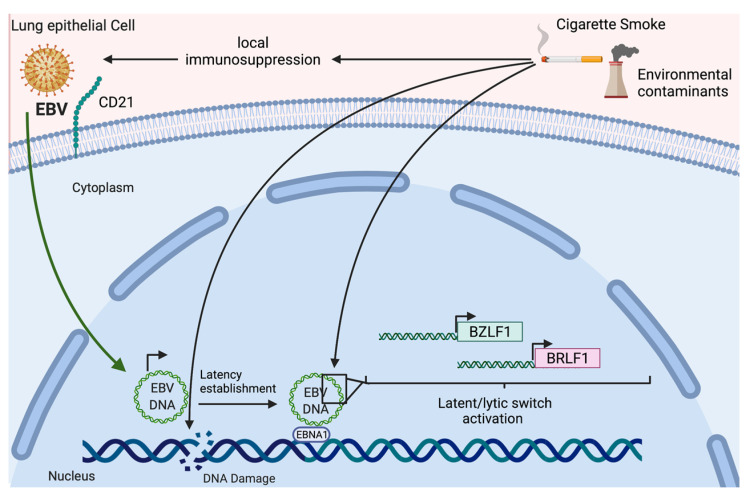
Hypothetical mechanism of interaction between cigarette smoke components or environmental contaminants with EBV in lung epithelial cells. This interaction involves local immunosuppression facilitating EBV infection, latency establishment, and latent/lytic switch activation.

**Table 1 pathogens-11-00132-t001:** EBV frequency in lung cancer.

Tumor Types	Total	EBV (+)	(%)	EBV Gene	Methods	Country	Ref
AdC, SQC, SCLC, LCC	80	5	6.3	EBNA	ISH, IHC, PCR	Japan	Kasai [145]
NSCLC	167	9	5.4	EBNA	ISH, Southern Blot, IHC	China	Wong [144]
AC, mesothelioma	130	0	0	Absent	ISH	United States	Conway [154]
SQC, AdC, LCC, LELC	127	11	8.7	EBNA	ISH, IH	China	Chen [137]
LELC, SQC, AdC, LCC, SCC	51	30	58.8	EBER, LMP1, VCA	ISH, IHC	China	Han [136]
SCC	23	1	4.3	LMP-1	ISH, IHC, PCR	United States	Chu [155]
SQC, AC, AdC-SCC, LCLC, SCLC	122	0	0	Absent	ISH, IHC, PCR	France	Brouchet [160]
NSCLC	108	36	33.3	EBER	ISH	China	Li [141]
SQC, AdC	19	12	63.2	EBER	ISH, PCR, IHC	Spain	Gomez-Roman [146]
AdC	110	0	0	Absent	ISH	Singapore	Lim [156]
SQC, AdC	48	7	14.6	BART1, BART2, and BHRF1–3	qPCR, Microarray	United States	Koshiol [157]
SQC, AdC, SCC	48	5	10.4	LMP-1	PCR	Iran	Jafarian [147]
NSCLC	66	4	6.1	LMP1 and EBNA1	NGS	China	Wang [151]
NSCLC	1017	3	0.3	EBNA-1, LMP-1 and LMP-2	NGS, ISH	United States	Kheir [152]
SQC, AdC, SCLC	73	2	2.7	Absent	PCR	Unites States	Gupta [148]
LELC	91	91	100.0	EBER	ISH-WES	China	Hong [138]
LC	108	36	33.3	EBER, LMP1, BCL-2	IHC	China	Li [142]
SQC	87	33	37.9	Absent	ISH	China	Zhang [143]
NSCLC, SQC	70	18	31.4	Absent	qPCR	Italy	Carpagnano [149]
LC	48	25	52	Absent	PCR, in situ PCR	China	Xia [150]
AdC	4	-	-	EBER	Microarray, qPCR, ISH	Korea	Kim [158]
NSCLC	Sample mix	-	-	Absent	Microarray, qPCR	Korea	Ma [159]

LC, lung cancer; NSCLC, non-small cell lung carcinoma; SQC, squamous cell carcinoma; AdC, adenocarcinoma; LCC, large cell carcinoma; SCLC, small cell lung carcinoma; PLELC, pulmonary lymphoepithelioma-like carcinoma.

## Data Availability

Not applicable.

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
