# Peer review of "Epstein–Barr Virus Infection in Lung Cancer: Insights and Perspectives"

_pathogens, 2022, doi:10.3390/pathogens11020132_

Round 1

Reviewer 1 Report

This is a revised version of the article made by Osorio and his group. Compared to the previous one, this version improves its weakness and adds some materials recommended by my opinions. However, this article still does not include enough information regarding the role of lytic proteins in lung cancer.

As the authors mentioned, nitrosamine, a tobacco component, induces EBV reactivation and tumorigenesis, which is a very logical direction to this issue. Moreover, a study you have cited, ref. 152, also reported that the BZLF1 is detected in NSCLC. Please introduce them more.

Author Response

Reviewer: This is a revised version of the article made by Osorio and his group. Compared to the previous one, this version improves its weakness and adds some materials recommended by my opinions. However, this article still does not include enough information regarding the role of lytic proteins in lung cancer.

As the authors mentioned, nitrosamine, a tobacco component, induces EBV reactivation and tumorigenesis, which is a very logical direction to this issue. Moreover, a study you have cited, ref. 152, also reported that the BZLF1 is detected in NSCLC. Please introduce them more.

Answer: Many thanks for this observation. An additional sentence was added in line 339-342.

Reviewer 2 Report

Osorio et al. present a very extensive review on EBV infection in lung cancer. They give a very thorough introduction both to the biology of Epstein -Barr virus as well as general ethiology of lung cancer.

The review encompases two large chapters, introduction and concluding remarks. The chapter referring to Epstein-Barr virus reviews current knowledge on virus characteristics and replication cycle, established role of EBV in epithelial cancers and role of EBV lytic and latent cycles in cancer. The second chapter discusses EBV infection in relation to lung cancer. In this part of the review the authors disscuss the following topics:

  1. Pathogenesis of lung cancer, including very thorough analysis of involvement of a long list of oncogenes in this process.
  2. Epidemiology of EBV in lung cancer - including a very helpful Table 1 summarizing frequency of EBV detection in lung cancer (comprising among others: used method, detected genes/proteins )
  3. Potential EBV-mediated mechanisms of lung carcinogenesis - thoroughly describing multiple mechanisms reported in literature.

General comments:

The manuscript is very well written and presents data in an organized fasion. It is a very careful and rigorous analysis of available data reviewing all currently known links between EBV infection and LC pathogenesis at the same time avoiding false conclusions. The text is supported by four figures and one table, which are generally helpful for understanding of presented concepts.

Specific comments:

  1. Figure 1 is helpful, however the size of the figure and the size of the font should be increased in order to improve legibility.
  2. The scheme presented in Figure 2 is excellent and summarizes the role of EBV in epithelial cancers in a very clear way.
  3. I am not sure if Figure 4 in the current format contributes in a significant way to understanding of the mechanism of lung cancer progression in presence of EBV. Perhaps it would be of an advantage to focus this scheme on the potential EBV-mediated lung cancer mechanisms related to interactions with tobacco components, a lot of information is available on this topic and the data for the remaining mechanisms are rather scarce.
  4. line 245: constitutive rather than constitute
  5. line 305: Recently, other type of EBV-related NSCLC was described, which display low....
  6. line 325: shouldn't it be : "in exhaled breath condensate"
  7. line 365: "the notion" should be removed
  8. line 366: I believe it should rather be "previous cell alterations are a required condition ( or pre-condition)"
  9. line 394: possible for tobacco smoke to affect...
  10. line 447: ... and can block the adaptive...
  11. line 457: These analyses suggest ...
  12. line 515: ...will help to elucidate... or ... will contribute to elucidation whether the infection occurs....

Author Response

Reviewer: Figure 1 is helpful, however the size of the figure and the size of the font should be increased in order to improve legibility.

Answer: The resolution of the image was improved.

Reviewer: The scheme presented in Figure 2 is excellent and summarizes the role of EBV in epithelial cancers in a very clear way.

Answer: Many thanks.

Reviewer: I am not sure if Figure 4 in the current format contributes in a significant way to understanding of the mechanism of lung cancer progression in presence of EBV. Perhaps it would be of an advantage to focus this scheme on the potential EBV-mediated lung cancer mechanisms related to interactions with tobacco components, a lot of information is available on this topic and the data for the remaining mechanisms are rather scarce.

Answer: Many thanks for this observation. The Figure 4 was changed, taking into consideration these comments.

Reviewer:

line 245: constitutive rather than constitute

line 305: Recently, other type of EBV-related NSCLC was described, which display low....

line 325: shouldn't it be : "in exhaled breath condensate"

line 365: "the notion" should be removed

line 366: I believe it should rather be "previous cell alterations are a required condition ( or pre-condition)"

line 394: possible for tobacco smoke to affect...

line 447: ... and can block the adaptive...

line 457: These analyses suggest ...

line 515: ...will help to elucidate... or ... will contribute to elucidation whether the infection occurs....

Answer: These corrections were included. Many thanks.

This manuscript is a resubmission of an earlier submission. The following is a list of the peer review reports and author responses from that submission.

Round 1

Reviewer 1 Report

This manuscript by Osorio et al. is a comprehensive, well-written review on the role of EBV in the pathogenesis of lung cancer. A few issues need to be addressed:

  • line 79, head of the section: "pathogenesis" instead of "pathogeny"; please check out other minor typos throughout the text
  • pages 5 to 7: the section dealing with the pathogenesis of lung cancer is too long, and need to be summarized
  • the authors cite a recent work on the same topic (10.3390/v13050877); they should also comment on the pathogenetic model proposed herein by Becnel et al., and eventually compare it with their own hypothesis 
  • lymphoepithelioma-like carcinoma of the lung is a distinct type of EBV-related lung cancer, therefore it should be discussed in more detail; likewise, other EBV-related lung tumors with different morphological features than LELC, yet similar molecular and epidemiological characteristics have been recently described (10.1097/PAS.000000000000117; 10.1097/PAS.0000000000001722), and deserve to be mentioned.

Reviewer 2 Report

This is a comprehensive review made by Osorio et al. to give insight and perspective on the EBV infection in lung cancer. The relationship between EBV and lung cancer is a very interesting issue.  The authors provide a very detailed summary in the background of EBV and pathogenesis of lung cancer "separately"; however thus, less information on EBV biology in lung cancer is given by this article. In addition, the section “3.3. Potential EBV-mediated lung cancer mechanisms” offers novel thinking in this issue, which will be much helpful for further studying. Some specific comments are listed below.

  1. The authors should strengthen the content regarding EBV biology or EBV status in different types of lung cancers.
  2. The oncogenic role of EBV lytic genes is paid more attention by EBV researchers. Several studies also detected the expression of lytic genes in lung cancers. Please introduce them.
  3. The cell biological studies of this topic should be included, not just epidemiological studies.

Reviewer 3 Report

Dear the authors,

The evidence that EBV infection is involved in the development of lung cancer is unclear. Therefore, the significance of this review paper is ambiguous. Even in the molecular biological studies you introduced, there are few studies using lung tissues and lung cells. Almost all data was from the studies by using BL LCL, NPC and GC cells. It is unknown whether the results of these studies can be applied to lung cancer.

  1. Has the monoclonality of EBV been proven in lung cancer?
  2. Do you know why different countries have different rates of EBV infection in lung cancer?
  3. Are there any reports of lytic gene expression in lung cancer tissue or surrounding non-cancerous tissue in vivo?
  4. There are many reports that EBV infection of epithelial cells is not dependent on CD21, but EBV infection is dependent on the expression of CD21 in the lung?